# SMYD3 Promotes Cell Cycle Progression by Inducing Cyclin D3 Transcription and Stabilizing the Cyclin D1 Protein in Medulloblastoma

**DOI:** 10.3390/cancers14071673

**Published:** 2022-03-25

**Authors:** Swapna Asuthkar, Sujatha Venkataraman, Janardhan Avilala, Katherine Shishido, Rajeev Vibhakar, Bethany Veo, Ian J. Purvis, Maheedhara R. Guda, Kiran K. Velpula

**Affiliations:** 1Department of Cancer Biology and Pharmacology, University of Illinois College of Medicine at Peoria, Peoria, IL 61605, USA; a.janardhan86@gmail.com (J.A.); kshido@uic.edu (K.S.); ianpurvis35@yahoo.com (I.J.P.); gmreddy@uic.edu (M.R.G.); velpula@uic.edu (K.K.V.); 2Department of Pediatrics, University of Colorado Anschutz Medical Campus, Aurora, CO 80045, USA; sujatha.venkataraman@cuanschutz.edu (S.V.); rajeev.vibhakar@cuanschutz.edu (R.V.); bethany.veo@cuanschutz.edu (B.V.); 3Department of Pediatrics, University of Illinois College of Medicine at Peoria, Peoria, IL 61605, USA; 4Department of Neurosurgery, University of Illinois College of Medicine at Peoria, Peoria, IL 61605, USA

**Keywords:** SMYD3, medulloblastoma, cyclin D1, cyclin D3, and cell cycle

## Abstract

**Simple Summary:**

Medulloblastoma is the most common malignant pediatric brain tumor and is classified into four molecular subgroups: Wnt, Shh, Group 3, and Group 4. Of these subgroups, patients with Myc+ Group 3 MB have the worst prognosis. Using an RNAi functional genomic screen, we identified the lysine methyltransferase SMYD3 as a crucial epigenetic regulator responsible for promoting Group 3 MB cell growth. We demonstrated that SMYD3 drives MB cell cycle progression by inducing cyclin D3 transcription and preventing cyclin D1 ubiquitination. Using in vitro and ex vivo studies, we showed that SMYD3 suppression by shRNA and BCI-121 significantly impaired proliferation, resulting in the downregulation of cyclin D3, cyclin D1, and pRBSer795. Moreover, we are the first to show that SMYD3 methylates the cyclin D1 protein, indicating that the SMYD3 stabilizes cyclin D1 through post-translational modification. Collectively, our studies position SMYD3 as a promising treatment option for Group 3 Myc+ MB patients.

**Abstract:**

Medulloblastoma (MB) is the most common malignant pediatric brain tumor. Maximum safe resection, postoperative craniospinal irradiation, and chemotherapy are the standard of care for MB patients. MB is classified into four subgroups: Shh, Wnt, Group 3, and Group 4. Of these subgroups, patients with Myc+ Group 3 MB have the worst prognosis, necessitating alternative therapies. There is increasing interest in targeting epigenetic modifiers for treating pediatric cancers, including MB. Using an RNAi functional genomic screen, we identified the lysine methyltransferase SMYD3, as a crucial epigenetic regulator that drives the growth of Group 3 Myc+ MB cells. We demonstrated that SMYD3 directly binds to the cyclin D3 promoter to activate its transcription. Further, SMYD3 depletion significantly reduced MB cell proliferation and led to the downregulation of cyclin D3, cyclin D1, pRBSer795, with concomitant upregulations in RB in vitro. Similar results were obtained following pharmacological inhibition of SMYD3 using BCI-121 ex vivo. SMYD3 knockdown also promoted cyclin D1 ubiquitination, indicating that SMYD3 plays a vital role in stabilizing the cyclin D1 protein. Collectively, our studies demonstrate that SMYD3 drives cell cycle progression in Group 3 Myc+ MB cells and that targeting SMYD3 has the potential to improve clinical outcomes for high-risk patients.

## 1. Introduction

Medulloblastoma (MB) is an aggressive embryonal brain tumor affecting the posterior fossa and is classified into four molecular subgroups: Shh, Wnt, Group 3, and Group 4 [1], with each subgroup displaying distinct genomic features, transcriptional profiles, clinical presentation, and prognosis [2]. Unfortunately, the classification of MB subgroups has not changed clinical practice; nearly all MB patients receive maximum safe resection, postoperative craniospinal irradiation, and chemotherapy [3]. Such aggressive treatments fail in a significant portion of patients with Group 3 MB and likely overtreat children with Wnt MB. Further, in light of significant advances in treatment regimens, standard therapies do not take into account the specific molecular mechanisms responsible for disease progression [4]. Of the four subgroups, Group 3 Myc+ MBs are molecularly less well characterized despite having the worst prognosis [5], necessitating the identification of the mechanisms responsible for Group 3 MB progression.

Epigenetic abnormalities resulting from defects in chromatin modifiers and remodelers often promote malignant transformation [6,7] and are commonly seen in MB [5,8]. Initial next-generation sequencing studies showed that mutations in epigenetic regulators account for the majority of genetic perturbations in Group 3 MBs [9,10], indicating that dysregulation of the epigenome is a crucial component in MB pathogenesis. Importantly, these epigenomic alterations are reversible and thus are attractive candidates for novel therapeutics to combat MB [11].

SMYD3 is a member of the SET and MYND-domain (SMYD) lysine methyltransferase family and plays an essential role in epigenetic regulation by acting as a transcription factor and targeting histones and non-histone molecules for methylation [12]. In this work, we identified SMYD3 as a critical epigenetic regulator of Group 3 Myc+ MB. SMYD3 promotes cell cycle progression by inducing cyclin D3 transcription via direct promoter binding, leading to retinoblastoma (RB) protein phosphorylation. SMYD3 inhibition using BCI-121 significantly reduced MB cell viability and was associated with the downregulation of Myc, cyclin D3, cyclin D1, and pRB in vitro and ex vivo. Additionally, we showed that SMYD3 stabilizes the cyclin D1 protein by preventing its ubiquitination. Collectively, these results position SMYD3 as an important therapeutic target in Group 3 Myc+ MBs.

## 2. Materials and Methods

### 2.1. Cell Lines and Transfection

The medulloblastoma cell line D283 MED was obtained from American Type Culture Collection (ATCC) and maintained in serum-containing advanced minimum essential medium (MEM). Normal human astrocytes (NHA), UW228, ONS-76, D425, D458, and MB002 cells (a kind gift from Dr. Rajeev Vibhakar, University of Colorado School of Medicine, Denver, CO, USA) were grown and transfected for 48 h using standard conditions [13,14]. All transfection experiments were performed with lipofectamine 2000 transfection reagent (ThermoFisher, Waltham, MA, USA). Briefly, the plasmids were mixed with lipofectamine reagent (1:2 ratio) in 200–500 µL of serum free-medium and left for 20 min at RT. The complex was then added to a 6-well or 100 mm plate containing 0.5 mL or 4 mL of serum-free medium, and around 1 µg plasmid per mL of medium was used. After 8 h of transfection, a complete medium containing 10% FBS was added and incubated for 48 h. Control cells were processed in the same way as treated cells and were incubated with either equal volumes of lipofectamine or transfected with vector alone.

### 2.2. Plasmids, Antibodies, Chemicals, and Inhibitors

The four unique GFP constructs of shSMYD3 were purchased from Origene (Rockville, MD, USA) and pXPG plasmid encoding a firefly luciferase reporter gene was purchased from Addgene (Cambridge, MA, USA). CRISPR/Cas9 knockout all-in-one plasmids for specific deletion of three non-overlapping regions of SMYD3 (KO_SMYD3 1/2/3) were purchased from GeneCopoeia, MD (HCP260717-CG04-3-B). SMYD3 overexpression plasmids were purchased from Sino Biologicals (Wayne, PA, USA). Anti-SMYD3 antibody was purchased from Abcam (Cambridge, MA, USA), cyclin D3 and p-RB antibodies were purchased from AB clonal (Woburn, MA, USA). HRP and Alexa-fluor conjugated secondary antibodies were purchased from Novus Biologicals (Littleton, CO, USA) and Invitrogen (Carlsbad, CA, USA), respectively. All the other antibodies were purchased from Santa Cruz Biotechnology (Santa Cruz, CA, USA). Human recombinant Cyclin D1/cdk4 and Cyclin D3/cdk4 protein complex was purchased from Thermo Fisher (Waltham, MA, USA).

### 2.3. In Silico Analysis

SMYD3 transcripts in MB tissues were assessed using the Oncomine database (Available online: http://www.oncomine.org (accessed on 2 July 2020)). The Kaplan–Meier survival curve was plotted using the Cavalli dataset within the R2 program (Available online: http://r2.amc.nl (25 March 2021)). Kyoto Encyclopedia of Genes and Genomes (KEGG) pathway annotations were used to identify gene sets significantly enriched for SMYD3 binding. The ‘Upstream Regulatory Analysis’ and ‘Pathway Analysis’ tools available within IPA software (Available online: http://www.ingenuity.com (28 January 2019)) were used to analyze the RNA-seq data.

### 2.4. Immunocytochemistry (ICC), Immunohistochemistry (IHC), Nuclear Extraction, Immunoprecipitation (IP), and Mass-Spectrometry

The ICC and IHC were done using a standard protocol [13,15]. Nuclear extracts were isolated using the Active motif kit (Carlsbad, CA, USA), IPed proteins were analyzed by LC-MS/MS spectra analysis at the UIC core facility, Carver Biotechnology Centre, Urbana, IL, USA [16] (see Appendix A).

### 2.5. Ubiquitination Assay

For the ubiquitination assay, whole cell lysates (800 μg protein/sample) were incubated with UbiCapture-Q Matrix (VWR International, Batavia, IL, USA) by gentle agitation at 4 °C overnight to pull down all ubiquitinated proteins according to the manufacturer’s instructions. After washing three times, captured proteins were eluted with 2× SDS-PAGE loading buffer and analyzed by western blotting using anti–cyclinD1 antibody as per the standard protocol [17,18].

### 2.6. Chromatin Immunoprecipitation (ChIP), Real-Time PCR, and RNA-Seq

ChIP assay was performed with the ChIP-IT Express Kit (Active Motif, Carlsbad, CA, USA) using standard protocol [19]. The anti-SMYD3 antibody ChIP-enriched DNA was sent for ChIP seq analysis (Arraystar Inc., Rockville, MD, USA). Total RNA was isolated using Qiatrizol reagent (Qiagen, Germantown, MD, USA) [13]. RNA sequencing was performed by the Biochemistry and Molecular Genetics core laboratory (University of Colorado, Denver, CO, USA) [20] (see Appendix A).

### 2.7. Cloning and Luciferase Reporter Gene Assay

The Cyclin D3 gene (CCND3) promoter region was analyzed for SMYD3 binding sites using the EPD database (available online: https://epd.epfl.ch//index.php (accessed on 24 June 2020)) and SMYD3_ChIP-seq data. The SMYD3 binding sites containing RI (ChIP identified region −260 to 300 bp) and RII (SMYD3 binding region within RI −277 to −7 bp) along with the RIII (SMYD3 negative control −800 to −400 bp) regions were amplified and cloned in pXPG luciferase reporter plasmid (Addgene, Watertown, MA, USA). The cloned plasmids were purified using GeneJET miniprep kit (Thermo Scientific) and subjected to restriction digestion and sequencing analysis (Eton Bioscience Inc., Union, NJ, USA). For the SMYD3 binding reporter assay, MB cells were transiently transfected with 1 μg/mL of the cloned cyclin D3 promoter (RI, RII, and RIII) plasmids. After 48 h, cells were lysed to determine luciferase activity using the luciferase reporter gene assay kit (BioAssay Systems, Hayward, CA, USA).

### 2.8. Cell Cycle, Annexin V Staining, TUNEL, Colony-Forming, and Cell Viability Assays

For cell cycle analysis, MB cells were washed with PBS, centrifuged at 870 g for 3 min at room temperature prior to fixation with 70% ethanol and treatment with RNAase (50 µg/mL stock). Cells were subsequently placed in propidium iodide (BioSure, Grass Valley, CA, USA) at 50 µg/mL concentration and incubated for 15 min at 4 °C. The proportion of cells in each phase of the cell cycle was assessed using Cell Quest software (Becton Dickinson Bioscience). To exclude the doublet population, the cells were sequentially gated then followed up with either an area-height or an area-width gate using forward scatter. The Annexin V assay was performed using the RayBio^®^ Annexin V450 apoptosis kit (RayBiotech, Corners, GA, USA) following the manufacturer’s instructions. TUNEL assay was done using an in situ cell death detection kit (Roche Applied Science, Indianapolis, IN, USA). The colony-forming assay was performed using control and shSMYD3 treated cells using an established protocol [21] (see Appendix A). Colonies were quantified by manual inspection and were graphically represented.

### 2.9. Slice Culture and Lactate Dehydrogenase (LDH) Detection

MED-411 FH, a patient-derived xenograft orthotropic mice tumor samples [14] (a kind gift from Dr. Rajeev Vibhakar, University of Colorado, Boulder, CO, USA) were dissected into 5 × 5 × 10 mm pieces and cut by a blade into 350 μm thick slices. Subsequently, slices were transferred onto cell culture inserts with a pore size of 0.8 μm (Millipore, Darmstadt, Germany) placed onto 24-well plates. A maximum of three slices were placed per insert and cultivated on a liquid-air-interface in a humidified incubator at 37 °C and 5% CO_2_. Each well contained 1 mL neurobasal culture medium under the membrane inserts supplying the tissue via diffusion. After 24 h, tissues were treated with BCI-121 (50 and 100 μM) for at least 72 h. To account for tumor-heterogeneity, cultivation was performed in triplets (*n* = 3) for each drug concentration and for the untreated control group. The culture media were changed every 24 h and slices were monitored for any signs of apoptosis. Media was collected on treatment days 0, 1, 2, and 3 for LDH analysis (G-Biosciences, MO, USA).

### 2.10. Statistical Analysis

Statistical analysis and graphing were performed using Origin version 9.0 software (Microcal Software Inc., Northampton, MA, USA). Image J software was used for all the densitometry and fluorescence intensity analyses. Statistical significance was calculated using one-way analysis of variance (ANOVA), and data were expressed as means ± S.E. In all Figures, statistical significance is labeled the following way: * *p* < 0.05, ** *p* < 0.01, *** *p* < 0.001.

## 3. Results

### 3.1. SMYD3 Is Crucial for MB Cell Survival

In collaboration with Dr. Vibhakar’s lab [11], we performed an RNAi-based functional genomic screen (Figure 1A) to identify which epigenetic regulators are involved in driving MB progression. The MB-derived Group 3 cell line D458 was transduced with a pooled lentiviral shRNA library targeting epigenetic regulators. The integrated epigenome-wide shRNA screen in D458 cells identified a cohort of genes that included SMYD3, as a key regulator in Group 3 MB cell survival (Figure 1B). Further, RT-PCR analysis confirmed that the Group 3 Myc+ MB cell lines (D283, D425, D458) exhibited higher SMYD3 transcripts when compared to normal human astrocytes (NHA) (Appendix A). SMYD3, though known in other cancers [22], is yet to be explored in MB, to our knowledge. To further characterize SMYD3 mRNA expression among the different MB subgroups, we analyzed the Shirsat and Pfister patient datasets, which revealed high expression of SMYD3, particularly in Group 3 MBs (Appendix A and Figure 2B). Immunohistochemical analysis further revealed that SMYD3 levels were significantly elevated in MB tissues (US Biomax, Inc., Derwood, MD, USA) when compared to normal cerebellum tissues (Figure 2A,B). The Kaplan–Meier survival curves revealed significantly poorer overall survival in Group 3 MB patients with high SMYD3 expression (*n* = 94) compared with patients with low SMYD3 expression (*n* = 19) (Figure 2C). These results demonstrate that SMYD3 is overexpressed in MB and that higher SMYD3 levels are associated with poor prognosis in Group 3 Myc+ MB patients.

### 3.2. SMYD3 Is Primarily Expressed in the Nuclei of MB Cells and Targets Cell Cycle Regulators

In addition to epigenetic regulation, SMYD3 can also influence signaling pathways outside of the nucleus [23]. Immunoblot analysis of cytoplasmic and nuclear extracts obtained from the D283, D425, and D458 cells revealed higher expression of SMYD3 protein in the nucleus (Figure 3A–C). To identify potential targets of SMYD3, the nuclear extracts from D283 cells were immunoprecipitated and the SMYD3-bound fractions were subsequently analyzed by mass spectrometry (Appendix A). In addition to histones and molecules involved in DNA binding, our analysis indicates that SMYD3 targets proteins involved in cell cycle regulation including CCND1, CCNB2, and KDM2A. These findings suggest that SMYD3 promotes MB cell survival possibly through protein-protein interactions and transcriptional regulation in the nucleus.

### 3.3. SMYD3 Transcriptionally Regulates Cell Cycle Effectors

To investigate the transcriptional impact of SMYD3, we performed ChIP-seq analysis to identify SMYD3-binding sites across the D458 genome. The heatmap of ChIP-seq vs. Input signals + 5 kb from the transcription start site (TSS) shows the 5′ ends of the start-seq reads are centered on the gene TSS, sorted by chromosome and position (Figure 4A). The sample peaks were annotated by the nearest gene TSS to the center of the peak region. After which, the peaks were divided into five classes based on their distances. A pie diagram was plotted to illustrate the distribution of the 9329 peaks based on the 5 classes delineated in this study (Figure 4B). These peaks represent the highest-affinity binding sites for SMYD3 and revealed SMYD3 enrichment within 5 kb of the TSS, indicating that SMYD3 may act as a distal regulator of gene transcription in MB cells. The ChIP peak at the global TSS appears lower than the Input peak (Figure 4C), suggesting that SMYD3 may play a greater role in transcriptional regulation outside of promoter binding. Further, SMYD3 appears to be enriched in the upstream and downstream regions of the TSS when compared to Input, implicating SMYD3 as a chromatin remodeler involved in both transcription initiation and elongation [24]. As our data indicate that SMYD3 promotes Group 3 MB cell survival through transcriptional regulation, we used the KEGG pathway annotations to determine which genes, in particular, were enriched for SMYD3 binding (Figure 4D). Notably, cell cycle genes including CCND3, CCNE1, and CDK2 were identified (Figure 4E). The cyclin E/CDK2 complex facilitates cell cycle progression [25], however, CDK2 alone is dispensable for cell proliferation [26]. Moreover, the expression and activity of cyclin E rely on cyclin D/CDK4/6-mediated phosphorylation of the retinoblastoma (RB) protein [27]. Thus, SMYD3 likely facilitates MB cell survival to a greater extent by influencing the expression of cyclin D3. Data mining using the Gilbertson MB patient dataset further revealed that of the four major MB subgroups, Group 3 and 4 MBs display elevated levels of cyclin D3 transcripts when compared to cyclin D1 (Appendix A and Figure 3B).

### 3.4. SMYD3 Induces Cyclin D3 Transcription

To determine whether SMYD3 affects the transcription of cyclin D3, we assessed cyclin D3 transcript levels by RT-PCR in the control and shSMYD3 treated Group 3 MB cells. SMYD3 knockdown was performed by transfecting MB cells with shRNA plasmids targeting SMYD3 (Appendix A). We found that SMYD3 knockdown significantly downregulated cyclin D3 mRNA (Appendix A) in Group 3 MB cells, in line with the ChIP-seq data. Using the eukaryotic promoter database, we detected four SMYD3 consensus binding sites distributed on the cyclin D3 gene promoter (Figure 5A). As the ChIP data revealed SMYD3 protein occupancy between the −311 and 367 bp region on the cyclin D3 gene promoter, we amplified this region and labeled it as RI. The RI site, which consists of two SMYD3 consensus binding sites was subsequently amplified, along with the RII (a subset of RI) and RIII regions (negative control) (Figure 5B). These regions were then cloned into a pXPG luciferase vector (Figure 5C) and confirmed by sequencing (Figure 5D and Appendix A). The reporter plasmids containing the cyclin D3 promoter regions RI, RII, and RIII were transfected into D458 cells, and the reporter activity of these plasmids was analyzed by luciferase reporter assays. The results suggest transcriptional activity in the RI and RII regions with the RI region exhibiting the highest luciferase activity (Appendix A). To validate whether SMYD3 transcriptionally upregulates cyclin D3, we transfected D458 and MB002 cells with the RI reporter plasmid in combination with shSMYD3, BCI-121 (SMYD3 inhibitor), or SMYD3 overexpression (SMYD3 OE) plasmid treatments. The shSMYD3 and BCI-121 treated cells showed reduced RI luciferase activity whereas SMYD3 OE augmented SMYD3 binding on the cyclin D3 gene promoter (Figure 5E,F). These results, along with the ChIP-seq data implies that SMYD3 directly controls the transcription of cyclin D3.

### 3.5. RNA Sequencing Supports the Role of SMYD3 in Regulating the Cell Cycle

To study the downstream effects of SMYD3 depletion in Group 3 Myc+ MB, we performed whole transcriptome analyses of shSMYD3 treated cells using RNA sequencing. In agreement with our previous data, the heatmap generated from the RNA-seq analysis revealed downregulation of CCND3 and CCNE1 and upregulation of p27, caspase 7, and caspase 9 (Figure 6A). To identify pathways altered by SMYD3 depletion, we performed a network analysis of the RNA-seq data. Ingenuity Pathway Analysis (IPA) revealed significant inhibition of cyclins and cell cycle regulation, mTOR, PI3K/AKT, and induction of apoptotic signaling (Appendix A). Moreover, we observed significant inhibition of CCND3, CDK4, Myc, and E2F3 transcriptional activity, with increased activity of the RB1 protein (Appendix A). To corroborate the RNA-seq data at the translational level, we performed immunoblot analysis on SMYD3 silenced Group 3 MB cells. The results confirmed that SMYD3 knockdown significantly reduced the expression of cyclin D3, Myc, CDK4, and pRBSer795 and increased the expression of the RB protein (Figure 6B,C). Interestingly, a significant decrease in cyclin D1 protein levels was detected, suggesting that SMYD3 also regulates the protein levels of cyclin D1.

### 3.6. SMYD3 Knockdown Is Associated with Increased Ubiquitination of Cyclin D1

The expression of cyclin D1 is predominantly influenced by its protein stability independent of de novo transcription [28]. For instance, Ras stabilizes the cyclin D1 protein by preventing its nuclear export and ubiquitination through AKT-mediated inhibition of GSK-3β [29]. O-GlcNAc transferase also stabilizes cyclin D1 in the nucleus through protein modification [30]. Our mass-spec analysis indicated that SMYD3 interacts with the cyclin D1 protein and identified two peptide sequences of cyclin D1 that are targeted for lysine methylation by SMYD3 (Appendix A, Figure 7A). Because SMYD3 is known to methylate lysine residues on MAP3K2 and AKT, leading to activation of Ras- and AKT-mediated signaling, respectively [23,31], we considered the possibility that SMYD3 may affect the protein stabilization of cyclin D1 via protein modification. To confirm SMYD3-cyclin D1 protein interaction, we immunoprecipitated (IP) the cell lysates from the GFP alone and GFP-tagged SMYD3 overexpressing D425 cells. The cell lysates were bound to the GFP-Trap^®^ Agarose beads tagged with GFP Nanobody (single domain GFP monoclonal antibody domain coupled to agarose beads). The IPed proteins were then immunoblotted for GFP, SMYD3, and cyclin D1 proteins (Figure 7B) as confirmed by the Z-stack co-localization studies (Figure 7C). Next, we performed a ubiquitination assay on D458 control and shSMYD3 treated cells to determine the impact of SMYD3 on cyclin D1 protein stability. We observed greater cyclin D1 protein levels in the ubicapture bound fractions in shSMYD3 treated cells when compared to controls (Figure 7D), indicating that cyclin D1 turnover is primarily regulated by ubiquitination in MB cells. Importantly, cyclin D1 ubiquitination was augmented in shSMYD3 treated cells when compared to controls, implying that SMYD3 stabilizes cyclin D1 by preventing its degradation. This study suggests that SMYD3-cyclin D1 interaction is associated with the increased stability of cyclin D1.

### 3.7. SMYD3 Knockdown Impairs Group 3 MB Cell Proliferation and Augments Apoptosis

We further validated the functional role of SMYD3 in MB cell proliferation by studying colony-forming ability in SMYD3 knockdown cell lines, D425 and D458. SMYD3 knockdown substantially reduced the number of colonies when compared to controls (*p* < 0.001) (Figure 8A,B). To investigate the effects of SMYD3 knockdown on the MB cell cycle, we performed flow cytometry analysis on D425 and D458 cells. Then, 24 h post-transfection, shSMYD3 treated cells showed a significantly greater number of cells in the G2/M phase when compared to controls (Appendix A). However, at 48 and 72 h post-transfection, the number of shSMYD3 treated cells in the G2/M phase was reduced whereas the number of cells in sub G1/apoptotic phase increased (Appendix A), suggesting that SMYD3 depletion inhibits cell growth by triggering cell cycle arrest followed by apoptosis. To confirm whether SMYD3 silencing induced apoptosis, we performed a TUNEL assay using an in situ cell death detection kit and found that shSMYD3 cells treated for 48 h displayed positive staining when compared to controls (Figure 8C,D). Likewise, increased Annexin V staining in shSMYD3 treated D425 and D458 cells confirmed that SMYD3 depletion increased the number of cells undergoing apoptosis when compared to controls (Figure 8E).

### 3.8. SMYD3 Regulates Downstream Effectors, Cyclin D3 and Cyclin D1, to Enhance MB Cell Viability

SMYD3 knockdown is associated with reduced Group 3 Myc+ MB cell survival and downregulation of cyclin D3/D1 proteins. To determine whether the addition of recombinant cyclin D1+CDK4 or cyclin D3+CDK4 complex proteins could antagonize the anti-proliferative effects of SMYD3 silencing, we performed an MTT assay. D458 cells were transduced with CRISPR plasmids to completely silence SMYD3 (KO-SMYD3) and the viability was compared with or without the addition of cyclin D1+CDK4 or cyclin D3+CDK4 complex. Predictably, MB cell viability was enhanced following the addition of cyclin D1+CDK4 or cyclin D3+CDK4 (Figure 9A). However, these effects were most prominent upon cyclin D3+CDK4 complex treatment. In contrast, cell viability was dramatically reduced following SMYD3 knockout. Moreover, the addition of cyclin D1 or cyclin D3 to KO-SMYD3 MB cells did not reverse the cytotoxic effects of SMYD3 depletion, suggesting that SMYD3 acts upstream of both cyclins. Next, we assessed the effects of cyclin D3 depletion in D458 and MB002 cells transfected with siRNA targeting cyclin D3 (siCCND3) (Appendix A) by flow cytometry. We found that a significantly greater percentage of siCCND3 cells were in the G0/G1 phase when compared to controls 24 h post-transfection, indicating a G0/G1 arrest. Although we observed that a substantial percentage of siCCND3 cells remained arrested in the G0/G1 phase 48 h post-transfection, the number of arrested cells was notably attenuated (Figure 9B,C and Appendix A), suggesting that a fraction of the siCCND3 cells initially arrested in G0/G1 were capable of returning to the proliferative pool.

It has been reported that cyclin D1 and D3 are critical for the growth of breast cancer cells in vitro and in vivo, with loss of cyclin D3 resulting in compensatory upregulation of cyclin D1 [32]. Therefore, we postulated that siCCND3 treated MB cells may be capable of continuous cell division through cyclin D1 upregulation. To address this, we examined the expression of both cyclin D1 and D3 in the nuclear extracts (NE) and cytoplasmic extracts (CE) of siCCND3 treated cells. Immunoblot analysis revealed barely detectable levels of cyclin D3 in both the NE and CE (Figure 9D). Importantly, the nuclear and cytoplasmic expression of cyclin D1 remained unchanged upon cyclin D3 depletion. Our findings are in line with a previous study in pancreatic adenocarcinoma, which found that CCND3 suppression did not result in compensatory upregulations in CCND1 expression [33]. Importantly, cytotoxicity and caspase 3/7 activity were significantly greater following SMYD3 knockdown compared to controls and siCCND3 treated cells (Appendix A and Figure 9B). Thus, the presence of cyclin D1 in the nuclei of siCCND3 treated cells would account for the aberration of cytotoxicity and caspase activity. As our data indicate that SMYD3 acts upstream to both cyclin D3 and cyclin D1, this further supports SMYD3 as an important therapeutic target of MB cell survival.

### 3.9. Pharmacological Targeting of SMYD3 Attenuates MB Proliferation Ex Vivo

The development of epigenetic therapies offers an alternative approach in treating recurrent tumors resistant to conventional therapeutics [34]. The small-molecule inhibitor BCI-121 selectively targets SMYD3-substrate interactions and has been extensively studied in other cancers [35,36,37]; however, its efficacy in MB has not been investigated. To assess the impact of BCI-121 in MB, we first performed an MTT assay. We found that BCI-121 impaired the viability of both D425 and D458 cells in a dose-dependent manner with an IC_50_ of ~80 μM, whereas the drug had no effect on normal human astrocytes (NHA) (Figure 10A). In addition to reduced SMYD3 expression, immunoblot analysis revealed that BCI-121 treatment led to the downregulation of pRBSer795 and upregulation of the RB protein (Appendix A). Although these results and numerous reports on BCI-121 highlight the potential benefits of targeting epigenetic modifiers for aggressive cancers, extensive intratumor heterogeneity poses significant challenges for assessing the success of anticancer therapies [38]. However, the use of patient-derived xenograft (PDX) models addresses some of these obstacles, as they retain several characteristics of in vivo tumor analysis, including intratumor heterogeneity [39]. As pharmacological inhibition of SMYD3 reduced MB cell proliferation in vitro, we sought to determine the efficacy of BCI-121 on MED-411 PDX derived MB tumors ex vivo. In this process, small sections of viable tumor tissues were subjected to short-term in vitro conditions to elicit a pharmacodynamic response (Figure 10B). We chose concentrations of 50 and 100 μM for the ex vivo assay based on the results obtained from the in vitro MTT assay. To estimate the degree of cytotoxicity induced by SMYD3 inhibition, we measured the release of lactate dehydrogenase (LDH). MED-411 tumors treated with 100 μM of BCI-121 showed noticeably greater LDH activity when compared to 50 μM of BCI-121 and the controls (Figure 10C). To confirm SMYD3 inhibition and assess the impact of BCI-121 treatment on cell cycle markers, tumor lysates were subjected to immunoblot analysis. At higher concentrations of BCI-121, the expression of SMYD3, cyclin D1/D3, and pRBSer795 were downregulated, whereas RB expression was upregulated (Figure 10D), mirroring the in vitro results. These data show that at higher concentrations, pharmacological inhibition of SMYD3 is cytotoxic to MB tumors ex vivo.

## 4. Discussion

Pediatric cancers, including MB, display a high degree of epigenetic dysregulation [40]. However, in contrast to genetic mutations, such epigenetic alterations are reversible [11] and are thus attractive targets for future drug designs. In addition to the six epigenetic drugs already approved for clinical use [41], multiple cancers have shown success with epigenetic therapy (NCT01752933, NCT01261312, NCT03189914), including histone deacetylase inhibitors and PI3K inhibitors for Myc-driven MBs [42]. In addition, the epigenetic modulators SETD8, BRD4, and BMI1 have been shown to be effective therapeutic targets for treating Group 3 Myc+ MB [11,43,44]. Here, we report an additional candidate, SMYD3, as a new promising target for epigenetic therapy against Myc amplified Group 3 MB.

Our studies demonstrate that SMYD3 inhibition attenuates the proliferative ability of Group 3 Myc+ MB cells and induces cell cycle arrest followed by apoptosis in vitro. Mechanistically, SMYD3 transcriptionally upregulates cyclin D3, resulting in RB phosphorylation and G1 phase progression (Figure 10E). Further, we showed that SMYD3 stabilizes the cyclin D1 protein in the nucleus by preventing its ubiquitination, likely through protein-protein interaction and lysine methylation.

The oncogenic properties of SMYD3 have been reported in numerous cancer types [45,46]. For example, in ovarian cancer, loss of SMYD3 results in the upregulation of CDKN2A (p16) and CDKN2B (p15) genes [35]. Similarly, we found that SMYD3 knockdown was also associated with increased transcriptional activity of p16, which may also be a consequence of Myc downregulation observed in SMYD3-silenced MB cells [47]. Furthermore, in breast cancer, SMYD3 drives cell cycle progression by upregulating the transcription of cyclin A1 [48] whereas, in hepatocellular carcinoma, SMYD3 promotes proliferation through stimulating CDK2 transcription [37]. Similarly, during carcinogenesis SMYD3 is recruited to the CCDN1 and CCNE1 promoters to activate their transcription [22]. Likewise, we identified SMYD3 enrichment at the CDK2 and CCNE1 genes, suggesting that SMYD3 also affects G1 phase progression by mediating the expression of the cyclin E/CDK2 complex. Moreover, SMYD3 knockdown was associated with the upregulation of p27, a potent inhibitor of cyclin D1/CDK4 and cyclin E/CDK2 complexes [49], providing another means by which SMYD3 mediates the cell cycle in MB. In both liver and colon tumors, SMYD3 interacts with RNA pol II to promote the transcription of genes involved in cell proliferation and metastasis [50]. Whether SMYD3-mediated effects on MB growth are dependent or independent RNA pol II [51] is yet to be established, however, uncovering such mechanisms will likely lead to the development of more effective therapeutics targeting SMYD3.

Both cyclin D1 and cyclin D3 are frequently upregulated in various cancers [52] and their aberrant expression is well documented in promoting oncogenesis [17]. In particular, cyclin D3 represents a valuable therapeutic target for its role in promoting cell cycle progression [53,54]. For instance, in pancreatic cancer cells, inhibition of cyclin D3 suppresses proliferation to a greater extent compared to cyclin D1 inhibition [33]. Our data revealed that when compared to cyclin D1, cyclin D3 expression is upregulated in high-grade Group 3 and 4 MBs and that SMYD3 occupies the cyclin D3 promoter to increase its transcriptional output. Furthermore, we found that inhibition of cyclin D3 only transiently arrested cells in the G0/G1 phase. Since cyclin D3 knockdown did not affect the nuclear expression of cyclin D1, this suggests that continuation of the cell cycle following cyclin D3 inhibition may be the result of persistent cyclin D1 expression [55]. Although we found that the levels of CCND1 transcripts were relatively low in Group 3 MBs when compared to other MB subgroups (Appendix A and Figure 3B), our studies indicate that SMYD3 interacts and stabilizes the cyclin D1 protein (Figure 7) in Group 3 MB cells. Therefore, in light of relatively low cyclin D1 transcripts when compared to other MB subgroups, Group 3 MB patients are more likely to show elevated levels of cyclin D1 protein relative to healthy individuals. Indeed, Group 3 MB cells/xenografts showed significant levels of cyclin D1 protein in Group 3 MB cells. On the other hand, RNA-seq analysis showed that SMYD3 knockdown significantly reduced the expression of Myc, MEK, and ERK, suggesting that SMYD3 may indirectly promote cyclin D1 transcription through the regulation of Myc, MEK/ERK signaling [56,57,58,59,60]. Nevertheless, the reduced expression of cyclin D3, cyclin D1, and pRSer795 and increased expression of the RB proteins supports the role of SMYD3 in driving cell cycle progression in Group 3 Myc+ MB cells and further validates transcriptional upregulation of cyclin D3 by SMYD3. Additionally, we showed that SMYD3 mediates the methylation of cyclin D1 and therefore likely influences cyclin D1 stability by preventing its ubiquitination. Understanding the exact mechanism of SMYD3-mediated ubiquitination will further establish the oncogenic functions of SMYD3 in MB. Because SMYD3 is capable of regulating cyclin D1 and cyclin D3 and likely controls the expression of cyclin E, targeting of G1 phase cyclins through SMYD3 inhibition offers a potentially more effective therapy for MB compared to targeting these cyclins independently.

An important consideration is the effect of BCI-121 on Group 3 Myc+ MB cells. BCI-121 directly targets SMYD3 to inhibit proliferation in cancer cells with high SMYD3 expression [36]. Studies have shown that BCI-121 induces S phase arrest and substantially represses tumor growth [15,33,34]. Similarly, we showed that BCI-121 reduced MB cell viability while SMYD3 knockdown led to G2/M arrest in vitro. Further, we demonstrated that at higher concentrations, BCI-121 was significantly cytotoxic to primary MB tumor samples with high Myc expression ex vivo and that SMYD3 knockdown significantly reduced Myc expression in vitro, further supporting the clinical benefits of targeting SMYD3 in Group 3 Myc+ MB patients.

Current treatment options for MB patients are relatively uniform, centered on craniospinal irradiation, chemotherapy, and palliative care [3]. As necessary as these therapies are, their clinical benefits are limited as they fail to capture the diverse genomic and transcriptional profiles unique to MB patients [61]. Given that patients with Group 3 MB commonly present with disseminated disease at the time of diagnosis [2], the need to develop specific therapies that target epigenetic modifiers such as SMYD3, is of utmost importance. Our findings further validate the significance of epigenetic regulators in MB progression and position SMYD3 as a promising treatment option for Group 3 Myc+ MB patients.

## 5. Conclusions

Using an RNAi functional genomic screen, we identified SMYD3 as an important epigenetic regulator of Myc+ Group 3 MB cell growth. We showed that SMYD3 drives MB cell cycle progression by activating cyclin D3 transcription, resulting in the phosphorylation of RB. We showed that SMYD3 inhibition significantly impaired MB cell proliferation while augmenting apoptosis. Mass-spectrometry analysis revealed that SMYD3 targets the cyclin D1 protein for methylation. As SMYD3 knockdown enhanced the ubiquitination of cyclin D1, our studies indicate that SMYD3 stabilizes cyclin D1 in the nucleus via lysine methylation. Ex vivo studies revealed that SMYD3 inhibition using BCI-121 was cytotoxic, further supporting the clinical significance of targeting SMYD3 in Group 3 Myc+ MBs.

## Figures and Tables

**Figure 1 cancers-14-01673-f001:**
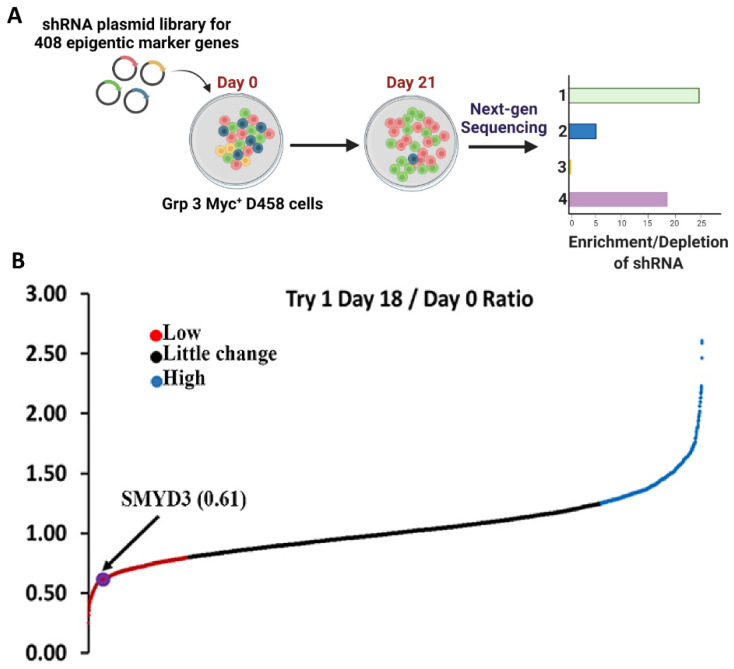
Functional genome-wide screening to identify novel epigenetic regulators in MB. (**A**) Schematics of the RNAi-based function genomic screen. In collaboration with Dr. Vibhakar’s lab [11], Pooled shRNAs targeting 408 genes (4–10 shRNAs per gene) involved in epigenetic regulation were transduced into D458 cells and analyzed for loss of function. Genomic DNA was isolated at 4 and 21 days after transduction, processed, and sequenced with an Illumina HiSeq instrument. The sequencing data compared the shRNAs on day 4 to day 21, with false discovery rates (FDRs) of 0.5 and 0.1, respectively. Underrepresented shRNAs are genes required for MB growth. (**B**) Aggregate normalized sequencing data showing depleted shRNAs (blue), unchanged shRNAs (red), and enriched shRNAs (green) at 21 days after transduction. The arrow shows the depletion of shRNA corresponding to SMYD3.

**Figure 2 cancers-14-01673-f002:**
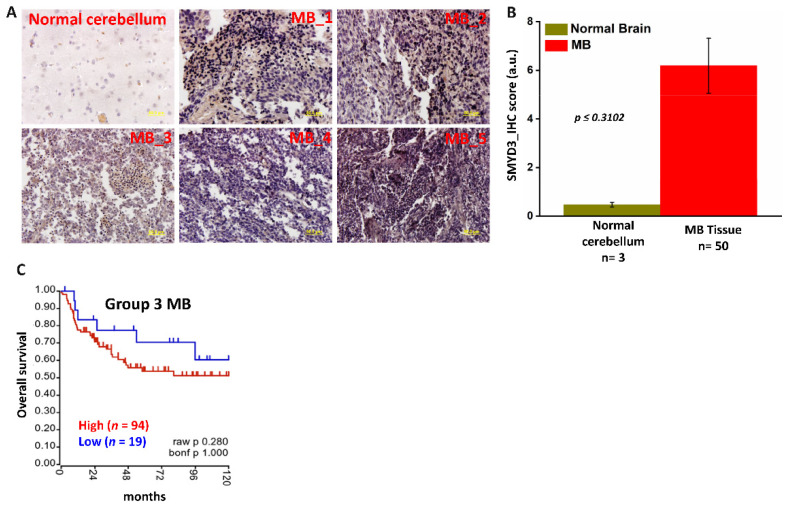
High SMYD3 expression is associated with reduced survival in MB. (**A**) Immunohistochemical (IHC) staining of normal human cerebellar tissue and MB tissues (magnification 40×; scale bar, 50 µm) (**B**) Q score analysis was performed using (staining intensity_Intden/no. of positive cells) ImageJ software to quantify SMYD3 protein expression. Staining for IHC analysis of MB (*n* = 50) and controls (*n* = 3) was performed per field (4×/per specimen) (**C**) Kaplan–Meier survival plot showing the overall survival probability of Group 3 MB patients with high (red) or low (blue) SMYD3 transcripts.

**Figure 3 cancers-14-01673-f003:**
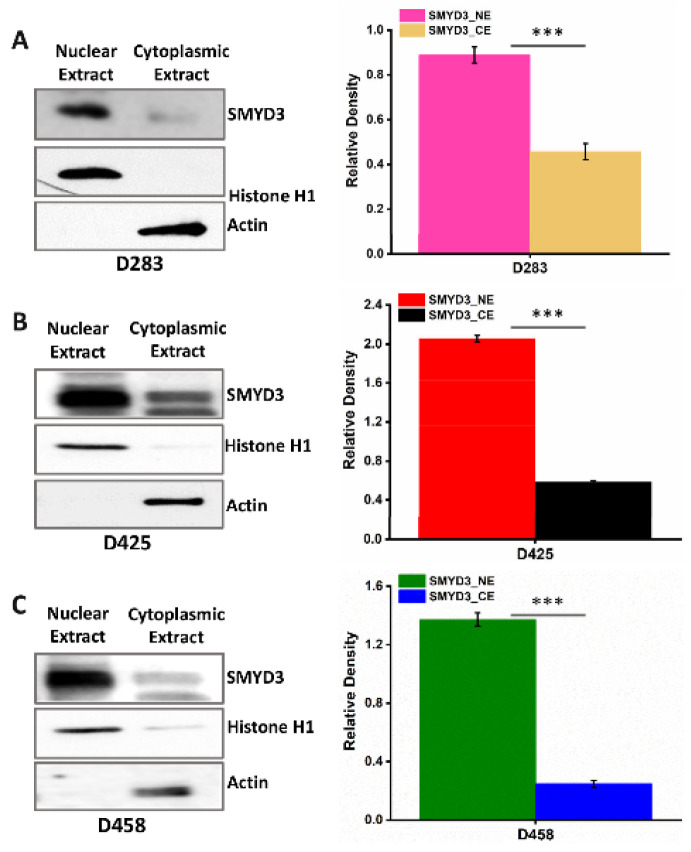
Group 3 MB cells express higher SMYD3 levels in the nucleus (**A**–**C**) Immunoblot of nuclear extracts (NE) and cytoplasmic extracts (CE) from D283, D425, and D458 MB cells, respectively using the anti-SMYD3 antibody. Densitometry analysis for the SMYD3 expression in the cellular fractions of Group 3 MB cell lines. The statistical significance *** *p* < 0.001.

**Figure 4 cancers-14-01673-f004:**
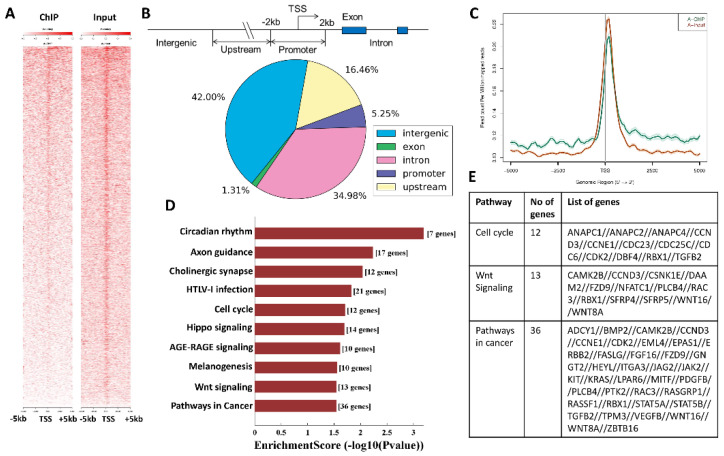
Chromatin immunoprecipitation (ChIP) analysis of transcriptomic and genome-wide SMYD3 binding profiles in MB. (**A**) Heat map showing the distribution and peak intensity of SMYD3 genomic occupancy from 5 kb downstream to 5 kb upstream. (**B**) Schematic representation of peaks into promoter peaks, upstream peaks, intron peaks, exon peaks, and intergenic peaks; pie diagram showing the distribution of peaks. (**C**) Graph depicting the ChIP peak at the global TSS (green) vs. Input peak (orange). (**D**) Pathway analysis using the Kyoto Encyclopedia of Genes and Genomes (KEGG) database showing SMYD3 enrichment scores. (**E**) List of genes and their representative pathways whose promoters were bound by SMYD3, as predicted by KEGG pathway analysis.

**Figure 5 cancers-14-01673-f005:**
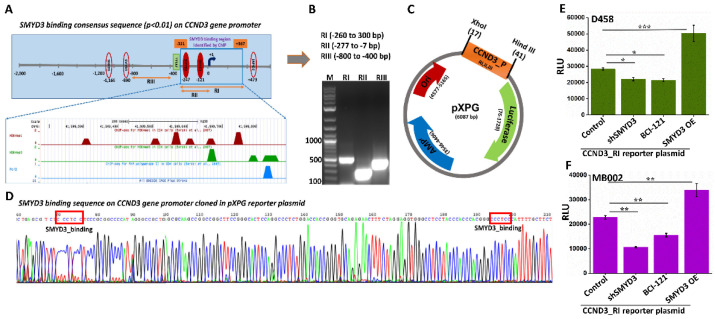
SMYD3 binds to the Cyclin D3 promoter. (**A**) Representation of the cyclin D3 (CCND3) gene promoter sequence; (red circles) the EPD software detected four SMYD3 consensus binding sites (−1166, −890, −247, −121 bps) on the cyclin D3 gene promoter. (Blue box) Region −311 to +367 was identified by SMYD3_ChIP-seq analysis. (**B**) Agarose gel showing the PCR amplification of RI (ChIP identified sequence), RII (SMYD3 binding sites within RI), and RIII regions (negative control). (**C**) Scheme depicting the RI, RII, and RIII regions cloned into the pXPG reporter vector. (**D**) Sequence peaks showing the SMYD3 binding sites on the cloned CCND3 promoter regions within the pXPG plasmid. (**E**,**F**) Effect of SMYD3 on Cyclin D3 (RI region) promoter activity by luciferase reporter assay in D458 (top) and MB002 (bottom) cells. D458 and MB002 cells were transfected with shRNA or overexpression vectors targeting SMYD3 for 48 h or treated with BCI-121 (80 μM) for 24 h. The statistical significance * *p* < 0.05, ** *p* < 0.01, *** *p* < 0.001.

**Figure 6 cancers-14-01673-f006:**
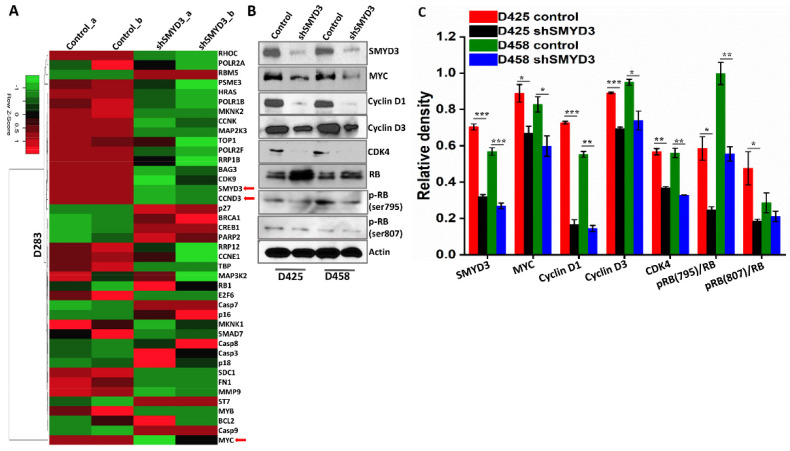
SMYD3 knockdown affects cell cycle molecules. (**A**) Heat map showing differentially expressed genes in D283 control vs. shSMYD3 transfected cells (48 h) in duplicates; upregulated genes are shown in red, downregulated genes are shown in green. (**B**) Protein expression and densitometric analysis of SMYD3, Myc, cyclin D1, cyclin D3, and CDK4 in the control and shSMYD3 treated D425 and D458 cells (48 h); actin was used as a loading control. (**C**) Graph depicting the densitometry analysis for the proteins in the western blot in MB cells. The ratio of the relative level of phosphorylated (*p*) RB to total RB is plotted for each experimental condition. The statistical significance * *p* < 0.05, ** *p* < 0.01, *** *p* < 0.001.

**Figure 7 cancers-14-01673-f007:**
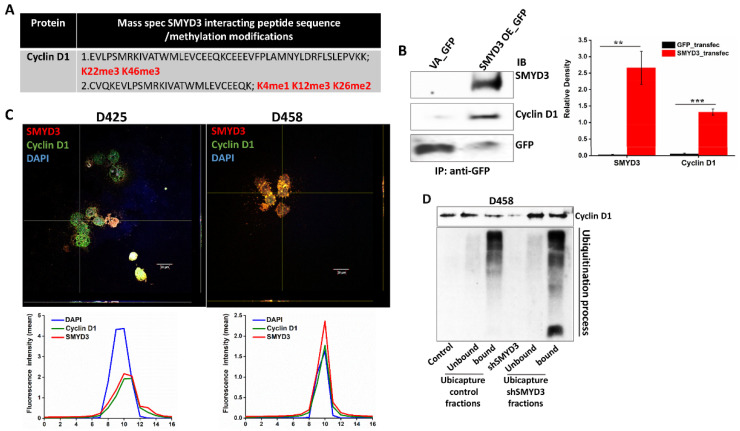
SMYD3-cyclin D1 interaction is associated with cyclin D1 stability in the nucleus. (**A**) LC/MS/MS analysis was performed using anti-SMYD3 antibody precipitated nuclear extracts of D283 cells. The mass spec analysis identified cyclin D1 peptide sequences and their lysine methylation modification sites. (**B**) D425 cell lysates (400 µg) from the GFP and GFP-tagged SMYD3 overexpressing cells were bound to the GFP-Trap^®^ Agarose beads (GFP Nanobody/single domain GFP monoclonal antibody domain coupled to agarose beads) for the IP of GFP-fused SMYD3 protein. The IPed proteins were then immunoblotted for GFP, SMYD3, and cyclin D1 proteins. The densitometry plot represents the ratio of the relative levels of SMYD3 or cyclin D1 to total GFP in each experimental condition. (**C**) Z-stack images (40× objective; 1-μm optical slice thickness) were captured using confocal microscopy (Olympus IX83) in MB cells to show the co-localization of SMYD3 (red) and cyclin D1 (green) proteins in the nucleus (DAPI). The fluorescence plot profiles in the lower panel confirmed that the two proteins, SMYD3, and cyclin D1 colocalize more in the nucleus (**D**) Ubiquitination assay on D458 control and shSMYD3 treated (48 h) cells. The cells were lysed and bound (40 µg) with a Ubicapture matrix to detect endogenous ubiquitinated cyclin D1 protein. The statistical significance ** *p <* 0.01, *** *p* < 0.001.

**Figure 8 cancers-14-01673-f008:**
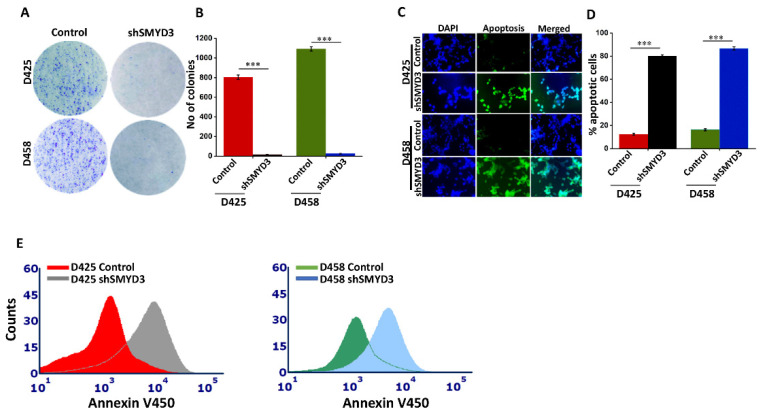
SMYD3 knockdown functionally impairs MB cell proliferation. (**A**) Clonogenic assay using SMYD3 silenced and control D425 and D458 cells. (**B**) Colonies were quantified manually and graphically represented in the control and SMYD3 silenced cells. (**C**) Tunnel assay showing the apoptotic cells (green) and the nucleus (blue) in D425 and D458 cells. The fluorescence intensity was quantified using Image J. (**D**) Bar graph showing the percentage of apoptotic cells in shSMYD3 treated cells, compared with the control cells. (**E**) Annexin V assay using the RayBio^®^ Annexin V450 apoptosis kit in the control and shSMYD3 treated D425 and D458 cells. The statistical significance *** *p* < 0.001.

**Figure 9 cancers-14-01673-f009:**
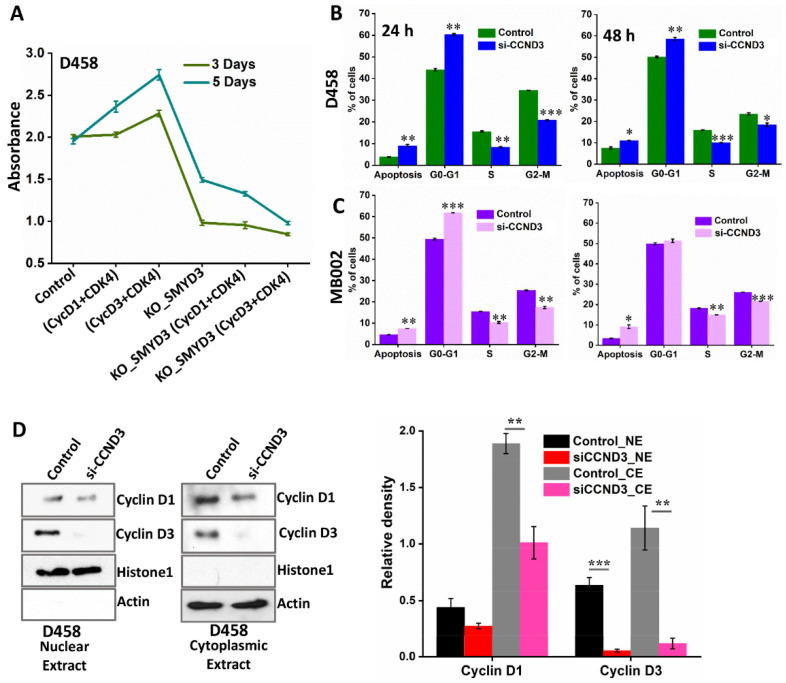
The effects of SMYD3 and cyclin D3 depletion on MB cell cytotoxicity. (**A**) MTT assay of D458 cells treated with cyclin D1/CDK4 or cyclin D3/CDK4 with or without SMYD3 knockout at 3 and 5 days. SMYD3 knockout (KO-SMYD3) CRISPR/Cas9 knockout all-in-one plasmids for specific deletion of three non-overlapping regions of SMYD3 (HCP260717-CG04-3-B) were used to treat D458 cells. (**B**,**C**) Bar graph depicting the percentage of control and siCCND3 treated D458 and MB002 cells in each phase of the cell cycle. (**D**) Western blot analysis of the nuclear extracts (NE) and cytoplasmic extracts (CE) obtained from the control and siCCND3 treated (48 h) D458 cells. Histone H1 and actin were used as the loading control for the NE and CE, respectively. Densitometry analysis for the cyclin D1 and D3 immunoblots of the NE and CE. The statistical significance * *p* < 0.05, ** *p* < 0.01, *** *p* < 0.001.

**Figure 10 cancers-14-01673-f010:**
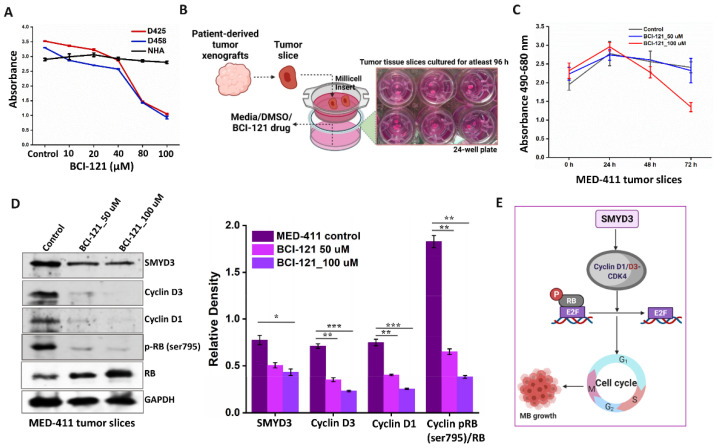
BCI-121 impairs MB cell viability in vitro and ex vivo. (**A**) MTT assay of D425, D458, and NHA cell lines treated with BCI-121. (**B**) Scheme showing patient-derived tumor xenografts (PDX) in the ex vivo tissue slice culture following treatment with BCI-121. (**C**) Graph depicting absorbance values (490–680 nm) of MED411-FH tumor tissue slices treated with or without BCI-121 over days 0, 1, 2, and 3. (**D**) Protein expression levels of SMYD3, cyclin D3/D1, and RB/pRB in the MED411 tumor tissue lysates treated with or without BCI-121 (50 and 100 µM). Graph depicting the densitometry analysis for the proteins in the western blot in the control and treated MED-411 tumor slices. The ratio of the relative level of phosphorylated (p) RB to total RB is plotted for each experimental condition. (**E**) Scheme representing SMYD3 regulation of cyclin D1/D3, its effect on RB phosphorylation, and progression of the cell cycle. The statistical significance * *p* < 0.05, ** *p* < 0.01, *** *p* < 0.001.

## Data Availability

The data will be made publicly available upon publication of this manuscript.

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
