# Peer review of "SMYD3 Promotes Cell Cycle Progression by Inducing Cyclin D3 Transcription and Stabilizing the Cyclin D1 Protein in Medulloblastoma"

_cancers, 2022, doi:10.3390/cancers14071673_

Round 1
Reviewer 1 Report
Dear Authors,
I am writing to express my review opinions on your manuscript titled SMYD3 promotes cell cycle progression by inducing cyclin D3 transcription and stabilizing the cyclin D1 protein in medulloblastoma.
Your study focused on studying the epigenetic regulation in group 3 medulloblastoma (MB) and characterized how SMYD3 regulates MB survival. Overall, the study is well-designed, the data are solid, and the writing is very good. There are the following main discoveries:
- Focusing on the epigenetic regulators, the authors conducted shRNAs screening and identified SMYD3 as essential for MB growth. High expression of SMYD3 in MB cancer and the correlation with poor survival rate further indicated that SMYD3 is important for MB survival. Most importantly, the authors clearly showed that pharmacological targeting of SMYD3 inhibits MB proliferation ex vivo.
- To identify SMYD3 interacting proteins, using co-IP followed by mass spectrometry analysis, they found that SMYD3 interacts with proteins that are involved in cell cycle regulation e.g. CCND1, CCNB2, KDM2A.
- To understand the chromatin binding profiling of SMYD3, using ChIP-seq, they found that SMYD3 acts as a distal regulator of gene transcription and it involves in both transcription initiation and elongation. More importantly, they found that SMYD3 directly binds to the TSS region of cell cycle regulators e.g. cyclin D3. Transcription of cyclin D3 induced by SMYD3 was also confirmed by reporter assays and qPCR.
- To identify genes that are regulated at the transcription level, using RNA-seq, the authors also identified cell cycle regulators as the downstream targets of SMYD3.
- Interestingly, the authors also found that SMYD3 stabilizes cyclin D1 by blocking its ubiquitination.
Detailed mechanism showing how SMYD3 regulates the cell cycle is novel. This manuscript also provides a deep understanding of how SMYD3 regulates gene expression in MB. This manuscript also meets the scope of “Cancers”. Only the following minor questions may be addressed, in order to publish:
- It has been reported that SMYD3 functions in the RNA polymerase II complex. So, is the role of mediating MB growth/survival dependent or independent of its function in acting as a component of RNA polymerase II complex?
- There are several studies showing that SMYD3 acts as a methyltransferase. Does SMYD3 also methylate proteins in MB? For example, could cyclin D1 be methylated by SMYD3?
- The study showed that SMYD3 blocks the ubiquitination of cyclin D1. How does SMYD3 perform this function?
- In figure 5D, the Co-IP result showing the interaction between SMYD3 and cyclin D1 is not very clear to me. Can you perform Co-IP using tagged SMYD3 and tagged cyclin D1, so the antibodies from different species can be used to avoid having a light chain and heavy chain in the blot?
We look forward to publishing your manuscript at Cancers after your minor revision.
Thanks
Author Response
March 10th, 2022
Editor-in-Chief
Cancers
RE: Manuscript ID: cancers-1595035 “SMYD3 promotes cell cycle progression by inducing cyclin D3 transcription and stabilizing the cyclin D1 protein in medulloblastoma”.
Dear Editor:
We would like to thank each reviewer for their comments and relevant suggestions. In this letter, we have responded to each concern raised by the reviewers and pointed towards adjustments made within the manuscript to address these suggestions appropriately.
Reviewer 1
Detailed mechanism showing how SMYD3 regulates the cell cycle is novel. This manuscript also provides a deep understanding of how SMYD3 regulates gene expression in MB. This manuscript also meets the scope of “Cancers”. Only the following minor questions may be addressed, in order to publish:
- It has been reported that SMYD3 functions in the RNA polymerase II complex. So, is the role of mediating MB growth/survival dependent or independent of its function in acting as a component of RNA polymerase II complex?
We would like to thank Reviewer 1 for making an insightful point regarding whether SMYD3-mediated MB growth is dependent or independent from the RNA pol II complex. Although such mechanisms were not investigated in this study, we have included this possibility into the manuscript (line 521-526).
- There are several studies showing that SMYD3 acts as a methyltransferase. Does SMYD3 also methylate proteins in MB? For example, could cyclin D1 be methylated by SMYD3?
We apologize to Reviewer 1 that SMYD3-mediated methylation of proteins was unclear. Our mass-spectrometry data was used to identify the SMYD3 associated protein peptides and their methylation modifications. The MS analysis of anti-SMYD3 antibody precipitated proteins identified that SMYD3 interacts with the cyclin D1 protein. Because the interaction between SMYD3 and cyclin D1 is associated with the methylation of cyclin D1 peptides, we decided to assess how SMYD3 expression can affect cyclin D1 expression and stability in Group3 MB cells. We made minor adjustments so that this point is represented correctly in the revised manuscript.
- The study showed that SMYD3 blocks the ubiquitination of cyclin D1. How does SMYD3 perform this function?
Reviewer 1 made an excellent point about the mechanisms of SMYD3-mediated ubiquitination of cyclin D1. Although our data show that SMYD3 mediates cyclin D1 methylation and stability in MB, SMYD3 knockdown was associated with cyclin D1 ubiquitination. The precise mechanisms have yet to be uncovered. As such, we have included some explanation (line 273-282) and also indicated this point in the revised manuscript (line 550-553).
- In figure 5D, the Co-IP result showing the interaction between SMYD3 and cyclin D1 is not very clear to me. Can you perform Co-IP using tagged SMYD3 and tagged cyclin D1, so the antibodies from different species can be used to avoid having a light chain and heavy chain in the blot?
Because showing the interaction between SMYD3 and cyclin D1 is important for our findings, as the reviewer suggested, we repeated the co-IP study using GFP tagged SMYD3 and cyclin D1 using antibodies from different species and is now included in Fig 7B. The MB cells expressing GFP tagged SMYD3 protein were lysed and the cell lysates were bound to the GFP-Trap® Agarose beads (GFP Nanobody/ single domain GFP monoclonal antibody domain coupled to agarose beads) for the IP of GFP-fused SMYD3 protein. This will avoid having a light chain and heavy chain in the blots. The IPed proteins were then immunoblotted for GFP, SMYD3 and cyclin D1 proteins.

Reviewer 2 Report
In the present study, the authors explored the role of SMYD3 in cell cycle progression in medulloblastoma. However, I have several major concerns as follows:
1- In figure S2A, densitometry analysis (SMYD3 expression normalized to actin) is essential to confirm the elevated expression of SMYD3 in Group 3 Myc+ MB cell lines as the SMYD3 expression in UW228 and ONS-76 looks almost like D458.
2- In Fig S2B, authors should also include mRNA expression data of UW228, and ONS-76 cell lines to compare the level of SMYD3.
3- In figure 2B, it is not clear whether the SMYD3 quantification for the IHC staining is performed per tissue specimen or per field for each tissue. Also, as an alternative to average density, Q Score analysis needs to be performed for the IHC quantification which takes into consideration for both the number of positive cells and also the staining intensity.
4- In Fig S3A, immunoblot analysis of cytoplasmic and nuclear extracts of different MB cell lines is essential for the confirmation of higher nuclear expression of SMYD3.
5- In Fig S3B, proper co-localization analysis (statistical analysis showing Pearson’s coefficient or intensity profile) is indeed essential to confirm the nuclear localization of SMYD3. Moreover, images seem saturated, while the merge images are fainter than single-channel images. The authors also failed to include scale bars. So, better quality images should be provided for figure S3b.
6- In Fig S4C and D, authors showed knockdown of SMYD3 in D283 cell line, however, the CyclinD3 mRNA level in D283 is missing in Fig S4d. Additionally, the authors didn’t provide the time point information or method of knockdown of SMYD3 (transfection or transduction).
7- In figure 4, the authors didn’t provide any information about the treatment duration of the SMYD3 inhibitor.
8- In Figure 5B, authors need to provide densitometric analysis of western blots and phospho to total protein ratio of pRB. Additionally, timepoint information is also missing.
9- In Figure 5D, the IP assay is not at all convincing with the high background of heavy chain and light chain. Furthermore, reverse IP is also missing to confirm the interaction.
10- In Figure S7, it is not clear from materials and methods and figure legend whether displayed images correspond to Z stacks. To demonstrate colocalization higher resolution Z stack images should be provided with proper statistical analysis.
11- In Figures 6C and D, the authors should mention the time points and description of the method of quantification for the tunnel assay. Additionally, the annexin V assay would strengthen the hypothesis of apoptosis induction.
12- Cell cycle analysis in Figure S8a, S8b, and Figure 7b: The method description of cell cycle analysis is incomplete. The authors didn’t provide any information about the fixation method and treatment with RNAse. The concentration of propidium iodide is too high which could be toxic to cells and also, they didn’t mention gating strategies to avoid doublets. Moreover, the citation for cell cycle protocol doesn’t match.
Furthermore, in figure S8a, S8b
- S phase (too small) and G2M (covering half of the S phase) gating is not appropriate in control and shSMYD3 (D458) at 24 hr and 72 hr.
- Go/G1 gating is not appropriate (too big) at 72 hr D425 and D458 and doesn’t match with other time points.
- Sub G1 peak is half visible and not suitable to draw any conclusion. Authors should repeat this cell cycle experiment in Group 3 Myc+ MB cell lines (D283, D425, D458) with proper voltage adjustment and gating of sub G1, G0/G1, S, and G2M phase.
- In Figure 7b, the authors should provide raw histogram profiles of cell analysis.
13- In Figures 6A and 6B, the Authors need to describe the quantification method for clonogenic assay.
14- In figure 7D, to confirm the purity of nuclear and cytoplasmic fractions, actin expression is needed in nuclear fraction and Histone H1 expression is needed in the cytoplasmic fraction. Moreover, authors need to provide densitometry analysis of all the biological replicates for further confirmation.
15- In addition to the PDX derived orthotopic MB tumors ex vivo culture, the pharmacological targeting of SMYD3 by BCI-121 needs to be validated in in vivo PDX model of medulloblastoma, either orthotopic or subcutaneous as shown by Sangar M.L.C et. el (PMID: 28637687).
16- In figure S9A, the actin blot is not clear. Authors should provide western blot image with proper actin expression along with densitometry analysis of SMYD3, Cyclin D1, CyclinD3 in Group 3 Myc+ MB cell lines.
17- In figure 8D, authors should include densitometry analysis of SMYD3, Cyclin D1, CyclinD3, p-RB, RB.
18- Authors must include western blot expression of cleaved caspases 3, 7, or 9 along with cleaved PARP-1 as markers for apoptosis induction with the knockdown of SMYD3 or CyclinD3.
19- In the supplementary figure section, the figure number needs to be included for all the supplementary figures.
20- In Figure S4A, the expression of CCND1 is low in group 3 in comparison to CCND3. However, the authors showed that SMYD3 also controls Cyclin D1 expression. A proper explanation needs to be addressed with experimental evidence.
21- There are significant format, typo, and grammar errors throughout the whole paper.
Author Response
March 10th, 2022
Editor-in-Chief
Cancers
RE: Manuscript ID: cancers-1595035 “SMYD3 promotes cell cycle progression by inducing cyclin D3 transcription and stabilizing the cyclin D1 protein in medulloblastoma”.
Dear Editor:
We would like to thank each reviewer for their comments and relevant suggestions. In this letter, we have responded to each concern raised by the reviewers and pointed towards adjustments made within the manuscript to address these suggestions appropriately.
Reviewer 2
In the present study, the authors explored the role of SMYD3 in cell cycle progression in medulloblastoma. However, I have several major concerns as follows:
- In figure S2A, densitometry analysis (SMYD3 expression normalized to actin) is essential to confirm the elevated expression of SMYD3 in Group 3 Myc+ MB cell lines as the SMYD3 expression in UW228 and ONS-76 looks almost like D458.
We agree with Reviewer 2 that densitometry analysis is essential for confirmation of elevated SMYD3 expression in the different MB cell lines. We have now included only the Group 3 MB cells as this study is more relevant to Group 3 MBs and added densitometry analysis for all the western blots performed.
- In Fig S2B, authors should also include mRNA expression data of UW228, and ONS-76 cell lines to compare the level of SMYD3.
Reviewer 2 has made a valid point that the mRNA levels of SMYD3 in UW228 and ONS-76 cell lines should be included given the comparison of SMYD3 protein expression. Since our study focused on Group 3 Myc+ MB, we have decided to only retain the expression data of Group 3 MB cell lines (SFig 1).
- In figure 2B, it is not clear whether the SMYD3 quantification for the IHC staining is performed per tissue specimen or per field for each tissue. Also, as an alternative to average density, Q Score analysis needs to be performed for the IHC quantification which takes into consideration for both the number of positive cells and also the staining intensity.
Thank you for this valuable comment. We have clarified in the results section that the IHC staining quantification was performed per field (4X/ specimen). We have made this correction and included a Q score to quantitate the number of positive cells and the staining intesnsity (Fig. 2B).
- In Fig S3A, immunoblot analysis of cytoplasmic and nuclear extracts of different MB cell lines is essential for the confirmation of higher nuclear expression of SMYD3.
We agree with Reviewer 2 that immunoblot analysis of the cytoplasmic and nuclear extracts in different MB cell lines would further support SMYD3 expression in the nucleus. We have included immunoblots for D283, D425, and D458 nuclear and cytoplasmic extracts, as seen in main Figure 3 in the revised manuscript.
- In Fig S3B, proper co-localization analysis (statistical analysis showing Pearson’s coefficient or intensity profile) is indeed essential to confirm the nuclear localization of SMYD3. Moreover, images seem saturated, while the merge images are fainter than single-channel images. The authors also failed to include scale bars. So, better quality images should be provided for figure S3b.
Reviewer 2 has made a good point. However, we decided to perform the cellular fractionation of various Group 3 MB cells to validate the nuclear localization of SMYD3 (Figure 3) as suggested in the previous comment. Our mass spec results also confirmed the presence of SMYD3 protein in the nuclear fraction. Because of this, we removed the immunocytochemistry data from the supplementary figures.
- In Fig S4C and D, authors showed knockdown of SMYD3 in D283 cell line, however, the CyclinD3 mRNA level in D283 is missing in Fig S4d. Additionally, the authors didn’t provide the time point information or method of knockdown of SMYD3 (transfection or transduction).
We have now included the cyclin D3 mRNA data for D283 (Figure S3D) into the manuscript. In addition to providing the time point information, the transfection was done using established protocol and the detail method is provided in the revised manuscript (line 81-88).
- In figure 4, the authors didn’t provide any information about the treatment duration of the SMYD3 inhibitor.
We apologize that the treatment duration for the D458 and MB002 cell lines was not addressed in the Figure legend. We have now included this information into the current manuscript (Figure 5).
- In Figure 5B, authors need to provide densitometric analysis of western blots and phospho to total protein ratio of pRB. Additionally, timepoint information is also missing.
The densitometry analysis for the western blots along with information regarding the timepoint is now included in Figure 6 in the revised manuscript.
- In Figure 5D, the IP assay is not at all convincing with the high background of heavy chain and light chain. Furthermore, reverse IP is also missing to confirm the interaction.
We agree with Reviewer 2 that the background of the IP assay is relatively high, making it difficult to clearly see the distinction between the heavy and light chain. As the reviewer suggested, we repeated the co-IP study using GFP tagged SMYD3 and cyclin D1 using antibodies from different species which is now included in Figure 7B. The MB cells expressing GFP tagged SMYD3 protein were lysed and the cell lysates were bound to the GFP-Trap® Agarose beads (GFP Nanobody/ single domain GFP monoclonal antibody domain coupled to agarose beads) for the IP of GFP-fused SMYD3 protein. This will avoid having a light chain and heavy chain in the blots. The IPed proteins were then immunoblotted for GFP, SMYD3, and cyclin D1 proteins.
- In Figure S7, it is not clear from materials and methods and figure legend whether displayed images correspond to Z stacks. To demonstrate colocalization higher resolution Z stack images should be provided with proper statistical analysis.
We have included Z stack images to confirm the co-localization of SMYD3-Cyclin D1 in the nucleus against DAPI, as seen in main Figure 7C in the revised manuscript. As the reviewer suggested, we also repeated the co-IP studies using GFP-tagged SMYD3. Together, the co-IP and the mass spec data confirm the interaction between SMYD3 and cyclin D1.
- In Figures 6C and D, the authors should mention the time points and description of the method of quantification for the tunnel assay. Additionally, the annexin V assay would strengthen the hypothesis of apoptosis induction.
We have included the time points and the methods in both the legends and Methods section. The figure has been changed to 8C and 8D in the current manuscript. Although the ApoLive assay measures viability, cytotoxicity, and caspase 3/7 activity (Figure S9A and S9B), we have included the Annexin V assay to validate apoptosis induction following SMYD3 knockdown (Figure 8E).
- Cell cycle analysis in Figure S8a, S8b, and Figure 7b: The method description of cell cycle analysis is incomplete. The authors didn’t provide any information about the fixation method and treatment with RNAse. The concentration of propidium iodide is too high which could be toxic to cells and also, they didn’t mention gating strategies to avoid doublets. Moreover, the citation for cell cycle protocol doesn’t match.
Furthermore, in figure S8a, S8b
S phase (too small) and G2M (covering half of the S phase) gating is not appropriate in control and shSMYD3 (D458) at 24 hr and 72 hr.
Go/G1 gating is not appropriate (too big) at 72 hr D425 and D458 and doesn’t match with other time points.
Sub G1 peak is half visible and not suitable to draw any conclusion. Authors should repeat this cell cycle experiment in Group 3 Myc+ MB cell lines (D283, D425, D458) with proper voltage adjustment and gating of sub G1, G0/G1, S, and G2M phase.
In Figure 7b, the authors should provide raw histogram profiles of cell analysis.
We apologize that Reviewer 2 thought our methods description for cell cycle analysis was insufficient. We have now detailed the exact protocol for the cell cycle analysis to better inform our readers. These changes can be seen in the methods section (line 143-149). We apologize for the typo in the propidium iodide concentration. The concentration of propidium iodide (50 µg/ml) was used as per the manufacturer’s guidelines and the concentrations established for the particular cell lines have been tested several times before and are now correctly reported in the revised manuscript. However, we did confirm this concentration with our in-house FACS analyst for reassurance. As the reviewer has suggested, we used a different software (De novo) to ensure proper gating and to avoid doublets. Please note, the FACS analysis were done at different time points (24, 48, and 72 h) after treatments and were compared with their respective control. However, we have elected to use the same gating template with the proper adjustments to more accurately highlight the differences among the sub G1, G0/G1, S, and G2/M phases for both the control and shSMYD3 treated cells at 24, 48 and 72 h.
The raw histogram FACS profiles of siCCND3 treated cells have also been included into Figure S8 in the revised manuscript.
- In Figures 6A and 6B, the Authors need to describe the quantification method for clonogenic assay.
We have now included the quantification method used for the clonogenic assay in the Figure 8 legends in the current revised manuscript.
- In figure 7D, to confirm the purity of nuclear and cytoplasmic fractions, actin expression is needed in nuclear fraction and Histone H1 expression is needed in the cytoplasmic fraction. Moreover, authors need to provide densitometry analysis of all the biological replicates for further confirmation.
In order to confirm the purity of the nuclear and cytoplasmic extracts, we have included actin and histone H1 expression in the immunoblots for the nuclear and cytoplasmic extracts, respectively. As the reviewer suggested, the densitometry analysis has been added for all the western blots.
- In addition to the PDX derived orthotopic MB tumors ex vivo culture, the pharmacological targeting of SMYD3 by BCI-121 needs to be validated in in vivo PDX model of medulloblastoma, either orthotopic or subcutaneous as shown by Sangar M.L.C et. el (PMID: 28637687).
We appreciate Reviewer 2’s enthusiasm for our project by expressing interest regarding the effects of BCI-121 in vivo. Our work with SMYD3 in MB is an ongoing endeavor; establishing the pharmacodynamics of BCI-121 is proposed in our future grant. Moreover, our IVIS machine is down, and due to COVID, there have been delays in the repair of the machine.
- In figure S9A, the actin blot is not clear. Authors should provide western blot image with proper actin expression along with densitometry analysis of SMYD3, Cyclin D1, CyclinD3 in Group 3 Myc+ MB cell lines.
We have replaced the actin blot and included proper densitometry analysis for SMYD3, cyclin D1, and cyclin D3 in the Group 3 Myc+ MB cell lines (Figure. S7).
- In figure 8D, authors should include densitometry analysis of SMYD3, Cyclin D1, CyclinD3, p-RB, RB.
We understand the importance of densitometry analysis for the western blots used in this study. As the reviewer suggested we have now included densitometry analysis and is included now as the Figure 10 D in the revised manuscript.
- Authors must include western blot expression of cleaved caspases 3, 7, or 9 along with cleaved PARP-1 as markers for apoptosis induction with the knockdown of SMYD3 or CyclinD3.
We agree with Reviewer 2 that an immunoblot of cleaved caspase 3/7/9 and cleaved PARP-1 would support apoptosis induction following SMYD3 suppression. The release of the enzyme lactate dehydrogenase occurs when cells lyse upon cell death. We showed that SMYD3 inhibition using BCI-121 was associated with greater levels and activity of LDH, indicating that SMYD3 inhibition resulted in apoptosis. In support of apoptosis induction, our lab was able to show that shSMYD3-treated cells had greater levels of caspase 3/7 activity (Figure S9) which we feel is superior to caspase 3/7 expression one would obtain from a traditional western blot. As Reviewer 2 suggested, we have performed Annexin V staining, a common method for detecting apoptotic cells.
- In the supplementary figure section, the figure number needs to be included for all the supplementary figures.
We have now included figure numbers for all supplementary figures.
- In Figure S4A, the expression of CCND1 is low in group 3 in comparison to CCND3. However, the authors showed that SMYD3 also controls Cyclin D1 expression. A proper explanation needs to be addressed with experimental evidence.
We would like to thank Reviewer 2 for the constructive comments and for mentioning the discrepancy regarding CCND1 and CCND3 expression in Figure S3A, and S3B in the current revised manuscript. Although analysis of one dataset of Group 3 MBs showed relatively low CCND1 expression when compared to other MB subgroups, this difference does not necessarily imply that SMYD3-mediated regulation of cyclin D1 expression is inconsequential. Like transcripts, proteins can be regulated at the post-translational level. Here, our study indicates that SMYD3 regulates cyclin D1 at the post-translational level via lysine methylation. Therefore, patients with Group 3 MBs are expected to have higher levels of cyclin D1 protein when compared to healthy individuals. Indeed, Group 3 MB cells/xenografts showed significant levels of cyclin D1 protein (Figure 6B, 6C, 7D, 9D, and 10D). Because this important point was unclear, we have included this in the discussion (line 537-547).
- There are significant format, typo, and grammar errors throughout the whole paper.
To ensure that our study has met all criteria for publication, we have sent our paper to be reviewed by our in-house linguist.

Reviewer 3 Report
The results presented by the authors are of interest in the search for new therapeutic targets for Group III MB. The authors claim that SMYD3 may act both at transcriptional level (modulating CyclinD3 expression) and at the posttranslational level (modulating CyclinD1 ubiquitination and degradation).
Interesting data are provided using large datasets (chipseq, RNAi screening, etc), less evidences are provided by wet molecular biology.
A few question remain unresolved:
- identification of the ubiquinating protein which targets CyclinD1 and demonstration of the effect of methylation on its binding, ubiquitination and degradation;
- More details on CyclinD3 promoter modulation: mutagenesis of the promoter, targeting SMYD3 binding sequences
Minor points:
correct line 185 (should it be...)
Figure 3A: the red colored names on the panels are not readable
CO-IP of SMYD3 and CyclinD1 is not of good quality.. the overall amount of IP protein is difficult to evaluate.
Author Response
March 10th, 2022
Editor-in-Chief
Cancers
RE: Manuscript ID: cancers-1595035 “SMYD3 promotes cell cycle progression by inducing cyclin D3 transcription and stabilizing the cyclin D1 protein in medulloblastoma”.
Dear Editor:
We would like to thank each reviewer for their comments and relevant suggestions. In this letter, we have responded to each concern raised by the reviewers and pointed towards adjustments made within the manuscript to address these suggestions appropriately.
Reviewer 3
The results presented by the authors are of interest in the search for new therapeutic targets for Group III MB. The authors claim that SMYD3 may act both at transcriptional level (modulating CyclinD3 expression) and at the posttranslational level (modulating CyclinD1 ubiquitination and degradation). Interesting data are provided using large datasets (chipseq, RNAi screening, etc), less evidences are provided by wet molecular biology. A few question remain unresolved:
- identification of the ubiquinating protein which targets CyclinD1 and demonstration of the effect of methylation on its binding, ubiquitination and degradation;
We would like to thank Reviewer 3 for their interest regarding the identification of the ubiquitinating protein responsible for cyclin D1 degradation. Although this is outside the scope of our study, we agree that identifying the ubiquitinating protein involved in cyclin D1 degradation is a worthwhile endeavor with significant implications for understanding MB progression. As such, we have included some explanation (line 273-282) and also indicated this point in the revised manuscript (line 550-553).
- More details on CyclinD3 promoter modulation: mutagenesis of the promoter, targeting SMYD3 binding sequences
We agree with Reviewer 3 that employing cyclin D3 promoter mutagenesis would augment our findings. We would kindly like to mention that the ChIP-seq data was confirmed by RT-PCR. The SMYD3 promoter binding was confirmed by luciferase reporter activity using the various clones (positive and negative) from the cyclin D3 promoter region. Further, we believe that the use of BCI-121 and shSMYD3-treated cells versus controls for the luciferase reporter assay is sufficient to conclude that SMYD3 induces cyclin D3 transcription. The use of mutagenesis at the cyclin D3 promoter would be ideal and necessary to demonstrate that SMYD3 activates cyclin D3 transcription, had we not performed this experiment with the aforementioned SMYD3 inhibitors. However, we will make sure we propose this in our ongoing endeavors and grant applications on SMYD3.
Minor points:
- correct line 185 (should it be...)
We apologize that we accidentally left a personal note within the Results section. It has since been removed.
- Figure 3A: the red colored names on the panels are not readable.
We were unaware that the words in red colour font (Fig 3A) could not be read. We have switched the font colours from red to black to improve the readability of the data.
- CO-IP of SMYD3 and CyclinD1 is not of good quality.. the overall amount of IP protein is difficult to evaluate.
Because showing the interaction between SMYD3 and cyclin D1 is important for our findings, as the reviewer suggested, we repeated the co-IP study using GFP tagged SMYD3 and cyclin D1 using antibodies from different species which is now included in Fig 7B. The MB cells expressing GFP tagged SMYD3 protein were lysed and the cell lysates were bound to the GFP-Trap® Agarose beads (GFP Nanobody/ single domain GFP monoclonal antibody domain coupled to agarose beads) for the IP of GFP-fused SMYD3 protein. This will avoid having a light chain and heavy chain in the blots. The IPed proteins were then immunoblotted for GFP, SMYD3, and cyclin D1 proteins. The density plots further confirm the results.

Round 2
Reviewer 2 Report
The authors have answered all the questions raised in the first round of review.
No further comment.